# STEP-INJECTION RECONSTRUCTION GUIDANCE FOR IMPROVING SINGLE ASPECT REAL IMAGE EDITING

## ABSTRACT

Latent Diffusion models have demonstrated the ability to generate realistic images, often derived from a text prompt. However, in many cases we have a pre-existing real image which we wish to change just one aspect of to generate the desired outcome – often referred to as single aspect image-to-image translation. There is no pre-existing tool which can perform this task directly, though people often build a pipeline which: i. generates both an image embedding and a prompt string which together would create an image as close as possible to the original image; ii. manipulating the prompt string to change the desired aspect – this could be done by substitution in the prompt string before mapping it to an embedding space or first mapping to an embedding space before manipulating this embedding; and iii. using the updated prompt embedding and the image embedding with the cross-attention mechanism, from a diffusion model, in an attempt to generate a new image which changes just one aspect of the original image. However, currently this type of approach often leads to multiple aspects of the original image being changed. To overcome this we propose the addition of a new step-injection reconstruction function applied to the early stages of the denoising process to provide additional guidance for final image construction. We demonstrate that our approach compares favorably to state-of-the-art results beating other approaches in terms of the DINO-ViT structure distance metric and arguably producing images which are closer to the original image save from the one aspect change that we desire. We go further to identify short-comings in two of the most commonly used metrics (Clip Accuracy and DINO-ViT structure distance) and propose two new metrics which allow for better evaluation and understanding of the results.

## 1 INTRODUCTION

Recent text-to-image diffusion models (e.g., Imagen (Saharia et al., 2022), DALL·E 2 (Ramesh et al., 2022), or Stable Diffusion (Rombach et al., 2022a)) trained on extremely large datasets are able to generate target images with guidance of a simple text prompt. The text prompt is first encoded by a text encoder to get a sequence of embeddings. Those embedding are then passed to cross-attention layers to give a semantic guidance during the image generation process, meaning that the produced image reflect the semantic meaning within the text prompt. However, we often wish to modify a real world image to change one aspect of the image without affecting the other aspects, for example changing an image of a cat sat on a red sofa to a dog (with the same gesture) sat on the same red sofa. This type of task (i.e., single aspect image-to-image translation) is not supported directly by diffusion models. Though this can be achieved by creating a pipeline which includes a pre-trained text-to-image latent diffusion model. First the pipeline would "reverse" the generation process to predict a text prompt and an image embedding which can be used together to reconstruct the "closest" image to the original one, then manipulate the prompt or its embedding before leveraging the cross-attention mechanism, in the text-to-image latent diffusion model, to produce the final image. However, there is no guarantee that changing part of text prompt would only change the desired aspect of the image without affecting other details of the original image.

Many approaches to image-to-image translation use a pre-trained latent diffusion model (most often stable diffusion model (Rombach et al., 2022a)) to retain, as similar as possible, the other features of the original image. Prompt-to-prompt (Hertz et al., 2022) extract cross-attention maps computed from the unedited text prompt, using them as a guide when generating the edited image. Pix2pix-

zero (Parmar et al., 2023) propose a pre-defined editing direction (e.g., an additional text embedding) for each type of transformation, along with the cross-attention guidance to generate edited images. Liu et al. (2024) extract self-attention maps, which capture the spatial information from the original image, using these maps when generating the new image from the edited prompt. Allowing them to maintain, to some extent, the structure and details of the original image. Park et al. (2023) use an energy function to improve the alignment between the edited text prompt and the image representation (e.g., feature maps) used to create the final image. This energy function is dynamically applied to update the prompts throughout the denoising process to better preserve the unchanged aspects from the original image. However, we would argue that the structure and unrelated aspects in the original images are still hard to retain in all of these approaches.

In our work, we adopt the approach in Pix2pix-zero of using a pre-determined direction for each type of transformation, thus making changes to the prompt more stable. Additionally during editing, we propose a step-injection reconstruction guidance along with the cross-attention guidance to better preserve aspects of the original image, providing better image-to-image translation compared to state-of-the-art models.

We identify limitations of the commonly used metrics (Clip Accuracy and DINO-ViT structure distance) and go further to proposed two new metrics derived from them aiming to provide more insight into how well the approaches work.

Our contributions are: i. We propose a step-injection reconstruction function during the early stages of the editing phase, showing remarkable improvement in terms of preserving the aspects of the image we do not wish to change. ii. We compare our method with the state-of-art methods, demonstrating the advantage of our approach. iii. We identify limitations of the commonly used metrics (Clip Accuracy and DINO-ViT structure distance), and, iv. We propose two new metrics (e.g., Clip Accuracy variation and Distance variation) to better evaluate and understand the results.

## 2 RELATED WORK

### 2.1 DIFFUSION MODEL

The training of the diffusion model can be defined by two processes: noising and denoising. The noising process keeps numerically adding noise to an image/image embedding, making it gradually converge to a simple distribution. While the denoising process uses a neural network that is trained to predict and remove the noise step by step to get back to the original image/image embedding. After training, the ideal model should have learned how to transfer random noise from a simple distribution to a realistic image.

The diffusion model originated from Sohl-Dickstein et al. (2015). In 2020, Denoising Diffusion Probabilistic Models (DDPM (Ho et al., 2020)) set the theoretical foundation for diffusion model training protocol. After this, extensive studies have been conducted to improve the performance of the diffusion model in various aspects. A paper published by Dhariwal & Nichol (2021) outperformed GANs on ImageNet (Deng et al., 2009) in 2021. Denoising diffusion implicit models (DDIM) (Song et al., 2020a) break the restriction of the Markov Chain, significantly reducing the required steps for sampling. Score-based Generative Modeling (SBGM (Song et al., 2020b)) provides more flexible sampling and improves the image quality by using stochastic differential equations (SDEs). Different from traditional diffusion models (Dhariwal & Nichol, 2021), (Sohl-Dickstein et al., 2015), (Song et al., 2020b), where the noising and denoising part are on the pixel level, latent diffusion model (e.g., Stable diffusion model (Rombach et al., 2022a)) uses an autoencoder to compress the image to a latent space, making the noising and denoising process happen in the latent space instead of image space. This approach significantly reduce the computational cost, making it a popular text-to-image model for image editing research. We build our pipeline using the Stable diffusion model.

### 2.2 IMAGE-TO-IMAGE TRANSLATION USING TEXT-TO-IMAGE LATENT DIFFUSION MODEL

A text-to-image latent diffusion model takes the latent representation (e.g., latent code) in the latent space and a text prompt as input to generate a text-conditioned image. Given a pre-trained stable diffusion model, many approaches explore its text embedding space and latent space to achieve

image-to-image translation. Many approaches leverage the cross-attention map (computed between embedding of the text prompt and latent representation of the image) and self-attention map (computed within the latent representation itself) to retain the unrelated aspects during editing. Prompt-to-Prompt (Hertz et al., 2022) proposes a "word swap" approach to achieve single aspect image editing (e.g., cat to dog, horse to zebra). They inject the cross-attention maps (obtained when generating the original image with the original prompt) into the editing generation process with the edited prompt. In this way, they retain the structure and other unrelated aspects of the original image. Pix2pix-zero (Parmar et al., 2023) shares the similar idea by using the original cross-attention map as a guidance to update the latent code, thus, achieving single aspect transformation. Liu et al. (2024) instead reuses self-attention map from the original image during denoising process and inject it into same process for producing the edited image. Park et al. (2023) propose an energy function to measure the alignment between the text prompt and the image representation. By using the guidance from the energy function to dynamically update the text context, the generated image is better aligned with the text prompt, performing more faithful transformation.

Although, these approaches alleviate the issue of poor performance of preserving other features from the original images, the performance still has its limits. The loss of detail and failure of retaining original image structures are still unsolved issues. In this work we apply an additional reconstruction guidance with the step-injection strategy to tackle these issues.

## 3 METHOD

### 3.1 OVERVIEW

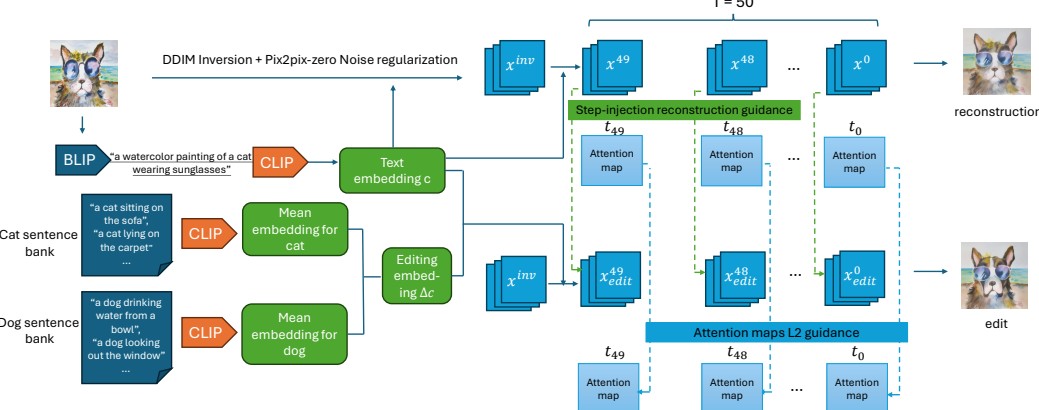

Figure 1: The overview for our approach. We first use DDIM inversion along with the Pix2pix-zero Noise regularization to get the inverse noise map $x^{inv}$, which is then used as the start point to reconstruct and edit the original images. During this process, the text embedding $c$ of the original image is computed by the CLIP model given the inferred text prompt predicted by BLIP model. We generate two sentence banks for cat and dog (for cat to dog transformation), which then are computed by CLIP model to get the text embedding. The editing embedding direction $\triangle c$ is obtained by taking the mean difference between cat and dog embedding. We combine two embeddings $c + \triangle c$ and feed into the generation process as the new text embedding during editing. The Attention map guidance and the Step-injection reconstruction guidance is applied to update the current latent noise map $x^t$, aiming to ensure the preservation of features from the original image.

The text-to-image diffusion model takes a latent code (e.g., noise map) and a text prompt as input to generate a text-conditioned image. Figure 1 illustrates our overall approach: Given a real image, we first invert the real image into a noise map in latent space which can be then used to reconstruct the image using a pre-trained diffusion model. We use the approach from Denoising Diffusion Implicit Models (Song et al., 2020a) (DDIM) with two regularization loss functions (the same as Pix2pix-zero (Parmar et al., 2023)) to obtain the noise map, further details in Sections 3.2 and 3.3. We employ BLIP (Li et al., 2022) to obtain the initial text prompt given a real image. To perform image-to-image

translation, we add a weighted pre-defined editing vector to the original text embedding, which is then used as the target prompt embedding to generate the edited image. However, We argue that only adding an editing direction to original text embedding is insufficient in terms of preserving the aspects of original image. Thus we apply cross-attention guidance and step-injection reconstruction guidance to preserve the aspects of the original image as much as possible.

We build our optimizations into the Pix2pix-zero model. In order to discuss these optimizations we first need to describe the core Pix2pix-zero model. This is presented in Sections 3.2 to 3.5.

## 3.2 DDIM INVERSION

Pix2pix-zero uses the DDIM inversion (Song et al., 2020a) to invert real images. Different from DDPM (Ho et al., 2020) where the forward and reverse process are stochastic, DDIM defines a non-Markovian, deterministic sampling process, making reconstructing a real image possible. Given a real image, the Variational Autoencoder (VAE) first encodes the original image to the latent representation (e.g., latent code) $x^0$. To reconstruct the original image, the diffusion model is used to denoise from $x^T$ to $x^0$, where $T$ denotes the total number of denoising timesteps. Therefore, the inversion process is to find the initial noise latent code $x^T$ which can be used to deterministically reproduce $x^0$ during the denoising process. Specifically, the U-Net denoiser $\epsilon_\theta(x^t, t, c)$ predicts the noise in the current latent code $x^t$ according to the current timestep $t$ and text embedding $c$, where $\theta$ are the parameters of the network. The estimated $\hat{x^0}$ can be obtained from equation 1, where the $\bar{\alpha}_t$ is the cumulative product of noise-schedule coefficients (Song et al., 2020a).

$$\hat{x^0} = f_\theta(x^t, t, c) = \frac{x^t - \sqrt{1 - \bar{\alpha}_t} \cdot \epsilon_\theta(x^t, t, c)}{\sqrt{\bar{\alpha}_t}} \tag{1}$$

The DDIM inversion uses equation 2 to iteratively map the original image latent code $x^0$ to the initial noise latent code $x^T = x^{inv}$.

$$x^{t+1} = \sqrt{\bar{\alpha}_{t+1}}\hat{x^0} + \sqrt{1 - \bar{\alpha}_{t+1}} \cdot \epsilon_\theta(x^t, t, c) \tag{2}$$

In this way, the $x^{inv}$ would be the latent code closest to Gaussian noise that can be used to reconstruct the original image.

## 3.3 PIX2PIX-ZERO (PARMAR ET AL., 2023) NOISE REGULARIZATION

Ideally, the predicted noise map $\epsilon_\theta(x^t, t, c)$ is expected to follow a gaussian distribution. It requires: i. There should be no correlation between any neighboring pixels in this noise map. ii. The noise map should have zero mean and unit variance. We apply the noise regularization during DDIM inversion (the same as Pix2pix-zero(Parmar et al., 2023)) to encourage the noise map to meet the above two requirements.

To achieve (i) we look at minimizing the pairwise loss between pixels. To reduce the computational cost of evaluating the correlation of all pairs given the initial noise map (shaped $64 \times 64 \times 4$) we apply a $2 \times 2$ average pooling to obtain the resolution of $32 \times 32 \times 4$, $16 \times 16 \times 4$ and $8 \times 8 \times 4$, forming a pyramid set$\{\eta_0, \eta_1, \eta_2, \eta_3\}$. Thus we seek to minimize $L_{pair}$

$$L_{\text{pair}} = \sum_p \frac{1}{S_p^2} \sum_{\delta=1}^{S_p-1} \sum_{x,y,c} \left( \eta_{x,y,c}^p \cdot \eta_{x-\delta,y,c}^p + \eta_{x,y,c}^p \cdot \eta_{x,y-\delta,c}^p \right) \tag{3}$$

where $\eta_{x,y,c}^p$ denotes the noise value at spatial location $(x, y)$ and channel $c$ at pyramid level $p$, where $S_p$ is the spatial resolution at that level. We use circular indexing to avoid boundary issues, and random values of $\delta$ are sampled at each iteration to capture long-range information.

To achieve (ii) we seek to minimize the KL divergence, thus making the noise as close as possible to a Gaussian distribution $\mathcal{N}(0, 1)$.

$$L_{\text{KL}} = \mu^2 + \sigma^2 - \log(\sigma^2 + \gamma) - 1 \tag{4}$$

where $\mu$ is the mean and $\sigma^2$ the variance and $\gamma$ a small constant to ensure numerical stability.

The final regularization loss is:

$$L_{\text{reg}} = L_{\text{pair}} + \lambda_{\text{KL}} L_{\text{KL}} \tag{5}$$

where the $\lambda_{\text{KL}}$ is the weight for KL loss.

### 3.4 Pre-defined Editing Direction in Text Embedding Space

Many image-to-image translation approaches choose to replace a word in the generated prompt text (referred to as word swap) to achieve image-to-image translation. However, this does not lead to a stable solution as the difference of only two words make it difficult to identify a good editing vector. Therefore, we use the same approach as Pix2pix-zero. First a thousand sentences, containing the word to be substituted, are generated using GPT-3.5 for example, "cat". Then another thousand sentences that contains the target word are generated, for example, "dog". The CLIP model (Radford et al., 2021) is used to embed these sentences in a text embedding space. The mean difference $\triangle c$ between these text embeddings is calculated and represents the editing vector between the substituted and target words (in our example "cat" to "dog"). This editing vector is added to the text embedding of the original image ($c_{ori}$) to produce the edited embedding:

$$c_{edit} = c_{ori} + s \cdot \triangle c \tag{6}$$

where $s$ is a scaler to adjust the strength of $\triangle c$.

### 3.5 Cross-Attention Guidance

To enable text-to-image generation, the stable diffusion model incorporates the cross-attention mechanism within its U-Net denoising network.

At each timestep $t$, the cross-attention layer computes Query (Q), Key (K) and Value (V) using the learned linear projections ($W_Q, W_K, W_V$):

$$Q = W_Q \cdot \phi(x^t), \quad K = W_K \cdot c, \quad V = W_V \cdot c \tag{7}$$

where $\phi()$ is the feature extractor within the U-Net network and $x^t$ comes from Equation 2. Here, the Query is obtained by projecting the intermediate spatial features $\phi(x^t)$ of the latent representation $x^t$, while the Key and Value are computed from the text embedding $c$. The cross-attention map is:

$$M = \text{Softmax}\left(\frac{QK^\top}{\sqrt{d}}\right) \tag{8}$$

where $d$ is the dimension of Q, K and V. The cross-attention map $M_{i,j}$ indicates the attention strength between $i_{th}$ location and $j_{th}$ text token. It encodes the relationship between generated image features and the text prompt, thus, containing the global layout and other details of the generated images.

We record the original attention map $M_{ori}$ and the attention map $M_{edit}$. We update the current latent code $x^t$ during denoising process by minimizing cross-attention loss:

$$\mathcal{L}_c = \left\| M^t_{edit} - M^t_{ori} \right\|^2 \tag{9}$$

Ideally, by encouraging the attention map $M^t_{edit}$ at timestep $t$ of the denoising process during editing to be close to the attention map $M_{ori}$ at $t$ during reconstructing, the features from the original image would be preserved.

### 3.6 Step-injection Reconstruction Guidance

We argue that cross-attention guidance is not enough to preserve the aspects of the original image during image-to-image translation – see results in Section 4.3. By constraining the similarity between edited latent code $X_{edit}$ and original latent code $X_{ori}$, the editing result would contain more information from the original image, presenting better aspect preservation during editing. Therefore, we propose an additional reconstruction guidance during the denoising process. In algorithm 1 we highlight the differences between the original Pix2pix-zero approach and our own. SAMPLING computes the next value of $x^t$, and was called UPDATE in the original work. We add extra inputs ($\lambda_r$, $\beta$) as the learning rate for the reconstruction guidance and the number of timesteps to apply our approach to. During Step 1 we additionally collect an array of the latent code for the next step ($x^{t-1}_{ori}$). Then during Step 3 we use this to compute our step-injection reconstruction guidance. Through experimentation we have identified that this is beneficial only when applied to the first $T - \beta$ timesteps. In order to move the latent code closer to the original, and hence the aspects closer

to the original, we compute the gradient for $x_{edit}^t$ based on $\mathcal{L}_r$ before updating $x_{edit}^t$ with this scaled by our learning rate $\lambda_r$, where

$$\mathcal{L}_r = MSE(x_{edit}^{t-1}, x_{ori}^{t-1}), \quad \text{for } t \in \{50, 49, \ldots, T - \beta\} \tag{10}$$

This, along with the edited vector $c_{edit}$ is fed through the diffusion model allowing us to predict the noise for this step. Finally this is fed through the SAMPLING function to recompute the next latent code which is guided towards the original image.

---

**Algorithm 1:** Algorithm for our editing pipeline

---

**Function** SAMPLING $(x^t, \hat{\epsilon}, t)$ :
     **return** $\sqrt{\alpha_{t-1}} \cdot x^t - \sqrt{1 - \alpha_t}\, \hat{\epsilon}/\sqrt{\alpha_t} + \sqrt{1 - \alpha_{t-1}}\, \hat{\epsilon}$;

**Input:**
     $x^T = x^{inv}$    the inverted latent code from DDIM inversion with noise regularization
     $c_{ori}$    the text embedding of the generated original prompt by BLIP
     $\triangle c$    the editing vector
     $\lambda_c$    the learning rate for cross-attention guidance loss $\mathcal{L}_c$
     $\lambda_r$    the learning rate for step-injection reconstruction guidance loss $\mathcal{L}_r$
     $\beta$    the step threshold for step-injection reconstruction guidance

**Output:** $x^0$ : the latent code for generating the final edited image

**Step 1:** ;      /* reconstruct the original image */
**for** *each reconstruction denoising timestep t = T...1* **do**
     $\hat{\epsilon}, \hat{M}_{ori}^t \leftarrow \epsilon_\theta(x^t, t, c)$ ;      /* $\hat{M}_{ori}^t$ is collected during denoising */
     $x_{ori}^{t-1} = \text{SAMPLING}(x_{ori}^t, \epsilon, t)$ ;      /* calculate $x_{t-1}$ for next iteration */
     $X_{ori} \leftarrow X_{ori} \cup \{x_{ori}^{t-1}\}$ ;      /* append $x_{ori}^{t-1}$ for reconstruction guidance */

**Step 2:** ;      /* calculate editing text embedding $c_{edit}$ */
$c_{edit} = c + \triangle c$
**Step 3:** ;      /* edit with cross-attention guidance */
**for** *each editing denoising timestep t = T...1* **do**
     $\hat{\epsilon}, \hat{M}_{edit}^t \leftarrow \epsilon_\theta(x^t, t, c_{edit})$ ;      /* $\hat{M}_{edit}^t$ is extracted during denoising */
     $\triangle x^t \leftarrow \nabla_{x^t} (\left\| M_t^{edit} - M_t^{ori} \right\|_2)$ ;      /* get gradient for $x^t$ by $\mathcal{L}_c$ */
     $x_{edit}^t \leftarrow x^t - \lambda_c \triangle x^t$ ;      /* update $x^t$ to get $x_{edit}^t$ */
     $\hat{\epsilon} \leftarrow \epsilon_\theta(x_{edit}^t, t, c_{edit})$ ;      /* recompute the $\hat{\epsilon}$ with updated $x_t$ */
     $x_{edit}^{t-1} = \text{SAMPLING}(x_{edit}^t, \hat{\epsilon}, t)$ ;      /* calculate $x_{edit}^{t-1}$ for next step */

     ;      /* apply step-injection reconstruction guidance */
     **if** $\beta \leq t \leq T$ **then**
         $\triangle x_{edit}^t \leftarrow \nabla_{x_{edit}^t} (MSE(x_{edit}^{t-1}, x_{ori}^{t-1}))$ ;    /* get gradient for $x_{edit}^t$ by $\mathcal{L}_r$ */
         $x_{edit}^t \leftarrow x_{edit}^t - \lambda_r \triangle x_{edit}^t$ ;      /* update $x_{edit}^t$ */
         $\hat{\epsilon} \leftarrow \epsilon_\theta(x_{edit}^t, t, c_{edit})$ ;      /* recompute the $\hat{\epsilon}$ with updated $x_{edit}^t$ */
         $x_{edit}^{t-1} = \text{SAMPLING}(x_{edit}^t, \hat{\epsilon}, t)$ ;    /* recompute $x_{edit}^t$ for next iteration */

---

## 3.7 QUANTITATIVE EVALUATION METRIC

### 3.7.1 CLIP ACCURACY VARIATION(CLIP ACC_V)

Previous approaches (Parmar et al., 2023; Park et al., 2023) use Clip Accuracy (Clip Acc) to evaluate how well the edited image align with the target prompt than the original prompt. The Clip Acc uses the CLIP model to embed the images and text into a joint embedding space. Then the portion of edited examples where the cosine similarity score between the edited image embedding and the target prompt embedding is higher than that between the edited image embedding and the original prompt embedding is computed as the metric. However, we argue that this binary evaluation metric has its limitations, since it has no indication of the magnitude of the differences between two similarity scores. Therefore, we propose a new metric named Clip Accuracy variation(Clip Acc_v) to

obtain a more insightful quantitative measurement.

$$Clipacc\_v = \frac{1}{N} \sum_{i=1}^{N} \frac{S_e^i}{S_o^i} \qquad (11)$$

where $S_e^i$ is the similarity score between edited image and target prompt embeddings, while $S_o^i$ denotes the score between edited image and original prompt embeddings. Giving a mean over the whole evaluation set, thus reflecting the magnitude of the differences between two scores.

### 3.7.2 DINO-ViT STRUCTURE DISTANCE VARIATION(DIST_V)

The Dist uses the deepest DINO-ViT (Caron et al., 2021) layer to extract keys from the attention module for each patch in the image, forming a key matrix. The matrix is then used to calculate the self-similarity matrix by comparing the key of each patch to the keys of all the other patches. The mean difference between two self-similarity matrices is used to indicate whether two images have the similar internal layout. However, we argue that the loss of details would not significantly affect this metric. For example, in the cat to dog transformation, if the mouth of the edited dog is missing or distorted, Dist will not punish this hard, since "missing/distorted mouth" does not strongly affect the internal relationship inside the image. Furthermore, the original metrics is not sensitive to the global layout because it focuses on relative relations within images.

We propose a variation for this metric named DINO-ViT structure distance variation (Dist_v). We extract the patch features from two images (e.g., original and target) before calculating the difference between each patch from source and target at the same position.

$$DIST\_v(I_1, I_2) = 1 - \frac{1}{N} \sum_{i=1}^{N} \frac{f_1^{(i)} \cdot f_2^{(i)}}{\|f_1^{(i)}\| \cdot \|f_2^{(i)}\|} \qquad (12)$$

where $f_1^{(i)}$ and $f_2^{(i)}$ denote the DINO-ViT embedding for the original image and target image, respectively. N is the number of patches. The metric returns a cosine similarity over all patches between two images, where a lower value indicates higher similarity. The propose Dist_v works directly on comparing the same position of two images, capturing the structural changes, spatial alignment as well as the content details between images.

## 4 EXPERIMENTS

### 4.1 IMPLEMENTATION DETAILS

We use a pre-trained Stable Diffusion v1.4 (Rombach et al., 2022b) for our experiments. We use GPT-3.5-turbo (OpenAI, 2023) to generate sentence banks. We present three transformations: i) cat to dog, ii) cat to cat wearing glasses, iii) horse to zebra, which are evaluated using common metrics (Clip Acc and Dist) as well as our new metrics (Clip Acc_v and Dist_v). Original images for evaluation of cat to dog and cat to cat wearing glasses are from the Laion-5b dataset (Schuhmann et al., 2022), whilst horse to zebra is from (louiscklaw, 2025).

For each transformation, we pick the best value $s$ for the text embedding $\triangle c$. Using $\{1.5, 1.5, 1.1\}$ for cat to dog, cat to cat wearing glasses and horse to zebra, respectively. For DDIM inversion, noise regularization is applied five times per timestep, using a weight of 20 for $\lambda_{KL}$. Applying 50 timesteps for inversion, reconstruction and editing processes. The classifier-free guidance (Ho & Salimans, 2022) is applied during editing. The guidance scale is set to 5 in all cases. We use a weight of one for $\mathcal{L}_c$ and a learning rate of 0.1; while for $\mathcal{L}_r$, the weight is $5 \times 10^4$ and the learning rate is 0.15. We use qualitative and quantitative (Clip Acc, Clip Acc_v, Dist and Dist_v) evaluation of our results.

### 4.2 QUANTITATIVE COMPARISON WITH STATE-OF-THE-ART MODELS

Table 1 show that we obtain the lowest value for Dist and Dist_v in cat to dog and horse to zebra transformations, presenting the best structural and detail preservation against other approaches. In cat to cat wearing glasses, we achieve the lowest Dist and the second lowest value in Dist_v while prompt-to-prompt performs best in Dist_v. However, our approach gives the lowest score in Clip

Table 1: Evaluation across three editing tasks with four metrics each. Best values are bolded.

| Method | (a) Cat → Dog | | | | (b) Cat → Cat w/ glasses | | | | (c) Horse → Zebra | | | |
|---|---|---|---|---|---|---|---|---|---|---|---|---|
| | CLIPacc↑ | CLIPacc_v↑ | Dist↓ | Dist_v↓ | CLIPacc↑ | CLIPacc_v↑ | Dist↓ | Dist_v↓ | CLIPacc↑ | CLIPacc_v↑ | Dist↓ | Dist_v↓ |
| DDIM + word swap | **98.2%** | **1.273** | 0.084 | 0.627 | **97.6%** | **1.206** | 0.082 | 0.619 | **98.7%** | **1.375** | 0.103 | 0.645 |
| prompt-to-prompt | 88.3% | 1.189 | 0.040 | 0.430 | 74.7% | 1.100 | 0.032 | **0.371** | 81.2% | 1.185 | 0.028 | 0.359 |
| Pix2pix-zero | 70.5% | 1.104 | 0.036 | 0.410 | 88.4% | 1.133 | 0.037 | 0.428 | 81.5% | 1.149 | 0.039 | 0.467 |
| Energy-Based | 81.5% | 1.145 | 0.038 | 0.453 | 96% | 1.194 | 0.047 | 0.54 | 94.6% | 1.271 | 0.044 | 0.501 |
| ours | 68.1% | 1.084 | **0.032** | **0.388** | 74.7% | 1.075 | **0.031** | 0.377 | 68.2% | 1.088 | **0.027** | **0.355** |

acc and Clip acc_v, suggesting that our approach produces results furthers from the target prompt. We argue that the Clip metrics are insufficient and inaccurate to indicate the quality as well as the accuracy of the transformations as shown in Section 4.4.

## 4.3 QUALITATIVE COMPARISON WITH STATE-OF-THE-ART MODELS

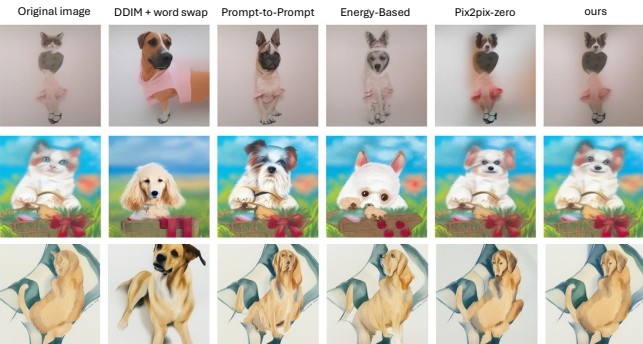

Figure 2: The real image editing on cat to dog transformation.

Figure 2 shows results for cat to dog transformations on real images. In the first example, DDIM + word swap, Prompt-to-prompt and Energy-based fail to preserve the pose of the original cat. Although Pix2pix-zero and ours both preserve most features of the original image, ours better preserves the ear shape, both still producing more of a dog face. The second example shows the similar case where the first three approaches have not successfully maintained the features from the original, while Pix2pix-zero and ours produce a dog image closer to the original cat. In terms of the detail preservation, ours keeps a better face shape as well as the head position than Pix2pix-zero. The third example is consistent with the others, ours achieves the best editing by keeping the gesture of the original cat and the details of the pillow (e.g.,the pattern of the pillow).

For cat to cat wearing glasses (Figure 3), ours shows the most natural editing seen in the first example (The glasses are not distorted). In the second example, the DDIM + word swap and Energy-based fail to preserve the details of the hat. Prompt-to-prompt although adding natural glasses to the cat,

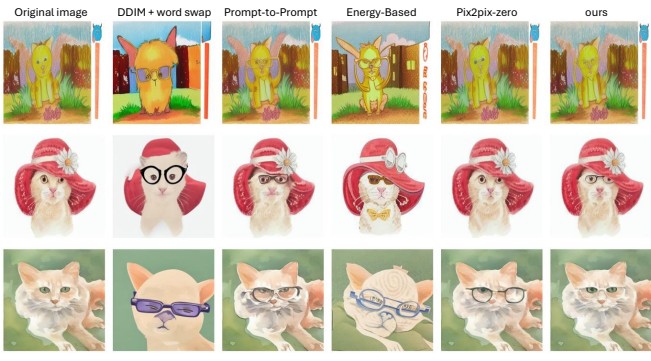

Figure 3: The real image editing on cat to cat wearing glasses transformation.

does not preserve the mouth detail well. Conversely, ours preserves the most details of the original image. Prompt-to-prompt and ours produce a successful transformation for the third example while the others failed in terms of fidelity.

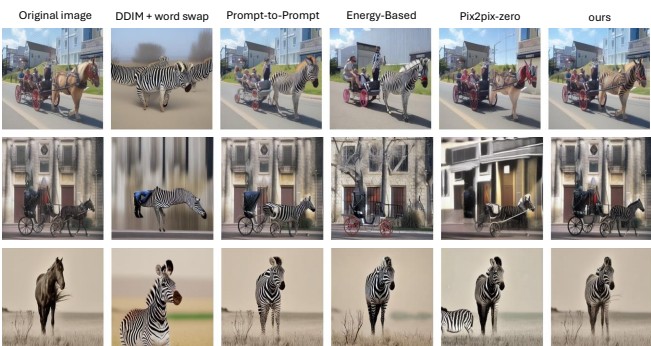

Figure 4: The real image editing on horse to zebra transformation.

Figure 4 shows the advantage of our approach in terms of preserving complex details in real images, especially in the first example and the second example, where ours restores the background best. In the third example, ours also retains the detail of the horse's tail, where others fail to keep this feature.

### 4.4 THE SHORTCOMING OF THE CLIP ACC METRIC

The measurement of the Clip Acc Metric indicates whether the edited image is closer to the target prompt than the original prompt and commonly used to indicate a better image translation. However, taking cat to dog transformation as an example, one can easily fool (maximize) the metric by pushing an image to extreme "dog" without caring for the similarity and fidelity of the image. To

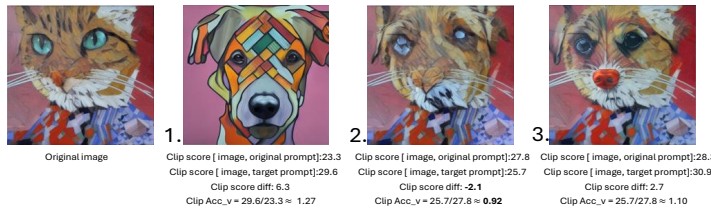

Figure 5: The example to illustrate the shortcoming of the Clip Acc metric.

exemplify this, we take the cat to dog transformation in Figure 5 which is edited using the different baselines and our approaches. We remove approach labels for the results to focus on the metric itself. Qualitatively one would rank the translations, best to worst as the third, second and first images. However, according to the Clip Acc metric only translations one and three are successes, even though image one shares little similarity with the original image. For the second translation, although distorted to some extent, it still successfully transforms the cat to dog and maintains similar texture and structure. Thus supporting our argument that Clip Acc metric is too crude if we wish to retain similarity between source and target image.

## 5 CONCLUSION

We propose the addition of a step-injection reconstruction guidance to the pipeline for image-to-image translation (e.g., cat to dog, horse to zebra) given a pre-trained stable diffusion model, aiming to better preserve the global layout and details of the original image while achieving a faithful editing. We compare favorably with state-of-the-art approaches, and present the advantages of our approach in terms of single aspect editing. Furthermore, we identify the limits of two commonly used evaluation metrics (Clip Accuracy and DINO-ViT structure distance) and propose two new metrics, aiming to providing better understanding and evaluation of the results.

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
