# OpenReview forum: "Step-injection reconstruction guidance for improving single aspect real image editing"
_ICLR.cc/2026/Conference — ICLR 2026 Conference Withdrawn Submission_

### Official Review · Reviewer_ohTQ · 2025-10-21

**Soundness:** 1
**Presentation:** 1
**Contribution:** 2
**Rating:** 2
**Confidence:** 5

**Summary:**

This paper presents a method to edit one aspect of a real image (*e.g.*, translating a cat to a dog) while preserving all other details by introducing a “Step-Injection Reconstruction Guidance “ mechanism within diffusion-based models. Building on Pix2pix-zero, the method combines DDIM inversion, cross-attention guidance, and a reconstruction loss applied in early denoising steps to anchor the edited image to its original structure. It also introduces two improved evaluation metrics to effectively alleviate the misleadings of widely-used metrics. Experimental results show that the proposed method improves structural fidelity and realism compared to the baselines.

**Strengths:**

S1. The paper recognizes the pitfalls of baselines on single-attribute manipulation tasks, and successfully alleviates the mentioned problem by introducing the step-injection reconstruction guidance.

**Weaknesses:**

**Major Weakness**

W1. The motivation of the task is weak. In the paper, the authors argue that “... we often wish to modify a real world image to change one aspect of the image …”, I understand that the authors would like to say editing an image “... without affecting other details of the original image.” I agree with this argument. However, I think there are many other cases that should be considered in text-driven image-to-image translation tasks, such as modifying multiple aspects, moving the object, enlarging/shrinking the object, and etc. Therefore I think the generalizability and applications of the proposed work is quite limited.

W2. Why do the authors not discuss the state-of-the-art image-to-image translation works? In the second paragraph of the introduction (Section 1), authors mention Prompt-to-Prompt [1], Pix2pix-Zero [2], InstructPix2pix [3], Liu et al. [4], and Park et al. [5]. However, there are more recent works that aim text-driven image-to-image translation.

W3. Related to W2: Authors say that “Many approaches to image-to-image translation use a pre-trained latent diffusion model …”. However, in recent years, many works leverage DiT [6] architecture for image editing framework. I think mentioning and discussing the works leveraging transformer architecture is also required. As discussed in Section 4.1., the authors use SD-v-1-4 checkpoint for experiment. I think experiments using recent backbone models (SDXL, or transfer-based diffusion models) should be held.

W4. Authors argue that they compare the proposed method with the SOTA methods (Both in Section 1 and Section 4), however, they compare with 1) DDIM [7], 2) Prompt-to-prompt [1], 3) Energy-based [5], and 4) Pix2pix-Zero [2], which is somewhat outdated (published in 2023). I think additional comparison with state-of-the-art works [8, 9, 10], including Nano Banana [11]-like recent works, is necessary.

W5. The novelty is limited. The authors proposed “step-injection reconstruction guidance”, however its design is too simple. In addition, the philosophy of minimizing the distance between source and target latent is already adapted in [12]. Also, many subsections of the method section (Section 3) discusses the preliminaries (Sec 3.1. ~ 3.5.), and only two subsections (Sec 3.6. ~ 3.7.) talks about novel components and some additional analysis. I think re-arranging the method section to emphasize the novelty of the work is also required.

W6. The experiment is very weak. Authors leverage only three tasks (cat -> dog and 2 additional tasks), which only contains a few comparisons. In addition, how many source image samples are leveraged per each translation?

**Minor Weakness**

W7. The overall writing and usage of the terminologies are fair; *e.g.* the expression ‘image embedding’ in the abstract is not familiar within the context; should it be referred as another wording involving ‘latent’ expression? In addition, the abstract is too long-winded, which can be written more shortly and compactly.

References

[1] Hertz, Amir, et al. "Prompt-to-prompt image editing with cross attention control." in ICLR (2023).

[2] Parmar, Gaurav, et al. "Zero-shot image-to-image translation." in SIGGRAPH (2023).

[3] Brooks, Tim, Aleksander Holynski, and Alexei A. Efros. "Instructpix2pix: Learning to follow image editing instructions." in CVPR (2023).

[4] Liu, Bingyan, et al. "Towards understanding cross and self-attention in stable diffusion for text-guided image editing." in CVPR (2024).

[5] Park, Geon Yeong, et al. "Energy-based cross attention for bayesian context update in text-to-image diffusion models." in NeurIPS (2023).

[6] Peebles, William, and Saining Xie. "Scalable diffusion models with transformers." in CVPR (2023).

[7] Song, Jiaming, Chenlin Meng, and Stefano Ermon. "Denoising diffusion implicit models." in ICLR (2021).

[8] Tang, Chuanming, et al. "Locinv: localization-aware inversion for text-guided image editing." in CVPRW (2024).

[9] Cao, Mingdeng, et al. "Masactrl: Tuning-free mutual self-attention control for consistent image synthesis and editing." in ICCV (2023).

[10] Lee, Hyunsoo, Minsoo Kang, and Bohyung Han. "Diffusion-based conditional image editing through optimized inference with guidance." in WACV (2025).

[11] Nano Banana, https://nanobanana.ai/

[12] Bar-Tal, Omer, et al. "Multidiffusion: Fusing diffusion paths for controlled image generation.", in ICML (2023).

**Questions:**

Please check the weakness section.

---

### Official Review · Reviewer_4Gze · 2025-10-29

**Soundness:** 2
**Presentation:** 3
**Contribution:** 1
**Rating:** 2
**Confidence:** 4

**Summary:**

This paper proposes an image editing method for text-to-image diffusion models. Building upon Pix2Pix-Zero, the authors introduce a step-injection reconstruction function that performs additional regularization during editing, preventing the edited image from deviating too far from the original one. This helps preserve the original image while applying the desired edits.

**Strengths:**

The paper is clearly written and easy to follow. The experimental setup is straightforward, and the proposed modification is simple to implement within existing diffusion-based editing frameworks.

**Weaknesses:**

**1) Limited Novelty**
The proposed step-injection reconstruction guidance is an incremental improvement over Pix2Pix-Zero. The core idea is to add a reconstruction constraint (MSE loss) between the edited latent and the original latent at early denoising steps, which is closely related to previously explored concepts in Null-text Inversion [1] and guided diffusion editing methods. Apart from restricting this loss to early timesteps, the paper does not provide substantial theoretical or algorithmic novelty.

**2) Incomplete Related Work Coverage**
The related work section is limited to early diffusion editing frameworks. However, the image editing landscape has evolved significantly, and the paper should also discuss more recent directions, such as methods that enhance inversion processes for more powerful editing and approaches based on SDS-guided optimization. The omission of these more recent directions weakens the paper’s positioning and makes it appear outdated relative to the current state of research.

**3) Weak Experimental Baselines and Limited Evaluation**
In the same context, the chosen baselines are relatively outdated and limited in scope. Editing tasks are overly simple and do not reflect the diversity or complexity of modern editing benchmarks. Moreover, the evaluation metrics rely only on CLIP and DINO-ViT scores, which fail to capture important aspects such as perceptual quality or human preference. Additional experiments on more challenging datasets and stronger baselines would be necessary to convincingly demonstrate the method’s advantage.

[1] Ron et al., *Null-text Inversion for Editing Real Images using Guided Diffusion Models*

**Questions:**

See the weakness.

---

### Official Review · Reviewer_XWvW · 2025-10-29

**Soundness:** 1
**Presentation:** 1
**Contribution:** 1
**Rating:** 2
**Confidence:** 5

**Summary:**

The paper introduces Step Injection Reconstruction Guidance, an image editing method that injects reconstruction terms at selected denoising steps to preserve structure and improve edit controllability.

My main concerns are that the paper is significantly outdated and the proposed approach is not practical. Specifically, it relies on Stable Diffusion 1.4, whereas state-of-the-art methods typically use FLUX, SD3.5, or at least SDXL. Moreover, the method requires additional per-image optimization, resulting in substantial computational overhead.
Overall, I recommend a negative evaluation of the paper.

**Strengths:**

* The paper considers the relevant problem of image editing.
* The paper discusses the drawbacks of current image editing metrics, which is meaningful.

**Weaknesses:**

* **Outdated methodology.** The approach is based on Stable Diffusion 1.4, while current state-of-the-art techniques use more advanced models such as FLUX or SDXL. As a result, the reported results are not competitive and do not demonstrate meaningful progress over existing work.

* **Limited practicality.** The method introduces an additional optimization step for each image, leading to significant computational overhead. This makes the approach impractical for real-world applications.

* **Limited evaluation and missing benchmarks.** The paper demonstrates only three editing cases without quantitative evaluation on standardized benchmarks. This raises concerns about the generalizability of the proposed method.

* **Poor writing.** The manuscript lacks clarity, and several sections are difficult to follow. Key design choices and motivations are not adequately explained, weakening the paper’s readability and impact.

**Questions:**

* What is the actual inference cost per image compared to standard diffusion editing methods?
* How does the choice or number of injection steps affect the quality of edits?
* Can the proposed method be integrated into few-step diffusion models (e.g., FLUX-Schnell) to improve efficiency?

---

### Official Review · Reviewer_SNeH · 2025-10-31

**Soundness:** 2
**Presentation:** 2
**Contribution:** 1
**Rating:** 2
**Confidence:** 5

**Summary:**

The paper proposes variation from pix2pix-zero for improved editing.

**Strengths:**

The proposed method improves structural consistency in editing process.

**Weaknesses:**

1. Baseline method of Pix2pix-zero is outdated (published in 2023), and the proposed method is only limited to marginal change from previous pix2pix-zero. Since there are thousands of recent editing methods, even in-context editing models, the practical contribution of proposed method is extremely suspicious.

2. Experimental settings are very weak. The paper shows comparison between traditional methods and only calculate CLIP-based metrics. The paper must include comparison between state-of-the-art methods such as FLUX-Context. Also there must be perceptual quality comparison such as human study.

3. The proposed method is NOT novel. DINO-based loss , MSE-matching between source latent, Cross-attention guidance, text embedding manipulation, all these methods are already proposed method and very famous. The paper proposed nothing new.

**Questions:**

No

---

### Note · Authors · 2025-12-22

I have read and agree with the venue's withdrawal policy on behalf of myself and my co-authors.